# Using the Internet: Nutrition Information-Seeking Behaviours of Lay People Enrolled in a Massive Online Nutrition Course

**DOI:** 10.3390/nu12030750

**Published:** 2020-03-12

**Authors:** Melissa Adamski, Helen Truby, Karen M. Klassen, Stephanie Cowan, Simone Gibson

**Affiliations:** The Department of Nutrition, Dietetics and Food, Monash University, Level 1 264 Ferntree Gully Road, Notting Hill, VIC 3168, Australia; melissa.adamski@monash.edu (M.A.); helen.truby@monash.edu (H.T.); karen.klassen@monash.edu (K.M.K.); stephanie.cowan@monash.edu (S.C.)

**Keywords:** nutrition education, information-seeking behaviour, nutrition misinformation, online learning, social media

## Abstract

People’s accessibility to nutrition information is now near universal due to internet access, and the information available varies in its scientific integrity and provider expertise. Understanding the information-seeking behaviours of the public is paramount for providing sound nutrition advice. This research aims to identify who learners in a nutrition-focused Massive Open Online Course (MOOC) turn to for nutrition information, and how they discuss the information they find. A multi-methods approach explored the information-seeking and sharing behaviours of MOOC learners. Summative content analysis, and an exploratory, inductive, qualitative approach analysed learners’ posts in MOOC discussion forums. From 476 posts, the majority (58.6%) of nutrition information sources learners reported were from websites. Providers of nutrition information were most commonly (34%) tertiary educated individuals lacking identifiable nutrition qualifications; 19% had no identifiable author information, and only 5% were from nutrition professionals. Qualitative themes identified that learners used nutrition information to learn, teach and share nutrition information. Consistent with connectivist learning theory, learners contributed their own sources of nutrition information to discussions, using their own knowledge networks to teach and share information. Nutrition professionals need to understand the principles of connectivist learning behaviours in order to effectively engage the public.

## 1. Introduction

Nutrition is a popular topic—there are umpteen websites, social media pages and books dedicated to food and eating, culinary skills, and the science of nutrition. Near-universal access to the internet, sophisticated search engines plus widespread adoption of social media now underpins the public’s access to information about food and health, and the public are increasingly turning to the internet for information [1]. However, the internet has no filters on quality or accuracy of information, enabling myths and pseudoscience to proliferate rapidly. People declaring themselves as ‘subject experts’ who lack formal training, reputable credentials or adherence to a professional code of conduct have a voice. Blogs, social media and ‘expert’ websites can be written by anyone, regardless of their expertise [2]. They can effectively influence their online followers and nutrition professionals need to acknowledge and compete in this space to be heard [3].

The emergence of technology that supports interaction and engagement with users, including the Web 2.0, with its emphasis on social networking and user-generated content, has aided nutrition and health information accessibility [4,5,6]. User-generated content refers to “media content created or produced by the general public rather than paid professionals, and is primarily distributed on the internet” [7]. People are increasingly accessing information from user-generated sources such as social media. In the United States, 80% of people reported seeking advice about their health online in 2011 and 70% reported social media as one of their sources of news [8,9,10]. Accessing food and nutrition information through social media is popular with the public, with it being the second most accessed area of science news, behind health and medicine [11]. Food, nutrition and health appear to be subject areas where people wish to share their personal experiences, beliefs and opinions, which are often either purported or interpreted as facts, rapidly propagating trends and fads. The challenge for the public is to be able to accurately differentiate between ‘fact or fad’ from the sheer volume of information available; the challenge for nutrition professionals is to deliver targeted messages in a manner to which the public will engage with them.

The shift from qualified nutrition experts disseminating the nutrition information they deem to be appropriate to the public taking charge of the information they learn has resulted in a paradigm shift in how nutrition is communicated [12]. Reliance and trust of evidence-based information presented by government and health care professionals with nutrition expertise has been reduced [13,14,15]. Internet users seeking health or medical information commonly favour commentary from people’s personal experiences on social media and blogs [9]. Celebrities and non-credentialed experts are frequently preferred as sources of credible nutrition information, ranging from movie and sports stars, to the new “health celebrities”, famous for their health and nutritional recommendations [16,17]. People not only look up to celebrities, but may be vulnerable to information provided by them, including misinformation [2,18,19].

As there are no restrictions as to whom provides online nutrition information, it produces a confusing world where consumers have to synthesise the ‘truth’ or indeed what meets their needs [20,21]. The resulting rise of user-generated content online combined with the public’s increased autonomy over their learning, makes understanding people’s behaviour around accessing nutrition information paramount. Tertiary qualifications in nutrition do not necessarily translate to ‘expertise’ in the eyes of the general public or make them receptive to receiving nutrition information [22].

Massive Open Online Courses (MOOCs) are an emerging free and open access source of education, enrolling millions of learners, and provide a platform for academics to reach global lay audiences [23,24]. A key aspect for MOOC learning is their discussion forums; allowing learners to converse, debate and share information with their peer group. These discussion forums can provide opportunities for observational research involving thousands of people engaging in natural dialogue, and these people can be viewed as representative of people actively seeking information online. Real-World Data (RWD), including information from discussion forums and social media, are increasingly recognised as useful in research to better understand the participant perspective [25]. Online discussion forums provide rich RWD on participants’ interactions and engagement with the course content and with each other. Observing people’s behaviour in sharing and reporting nutrition information provides a naturalistic window where we can further understand the characteristics of those seeking nutrition information and from where that information is being acquired. This is important to understand whether nutrition professionals are to be able to effectively deliver nutrition information using online formats.

While the internet and social media have been shown to be growing sources of nutrition information, less is known about who the providers of this information are. There are also gaps in understanding regarding who the public prefer to access online, and how they use the information they find. In order to provide timely nutrition advice that resonates with the public, nutrition professionals need to acknowledge and understand the rapidly evolving information-seeking landscape if experts are to compete in this fast-moving space.

This research harnesses data generated from online discussion forums from a nutrition-focused MOOC [26]. It aimed to identify the sources of nutrition information learners share with each other, to further categorise the source (providers’) vocational backgrounds using RWD. Synthesising the data generated by users of a nutrition-focused MOOC provides a rich and contemporary picture of where people who are actively seeking nutrition information are accessing that information and how they utilize the information they acquire and share.

## 2. Materials and Methods

A multi-methods approach was utilized to explore the information-seeking behaviours of MOOC participants. This included (i) a summative content analysis to categorise the sources of nutrition information and the vocational backgrounds of its providers from online conversations [27]; and (ii) an exploratory, inductive, qualitative approach to analyse learners’ posts in discussion forums throughout the MOOC, to explore how people used information they found on social media.

### 2.1. Data Collection

Nutrition professionals, who were academics from Monash University, designed, developed and delivered a three-week evidence-based, nutrition-focused MOOC called ‘Food as Medicine,’ which aimed to provide contemporary learning based on the Australian Healthy Eating Guidelines with a whole of diet approach to supporting nutritional health [28].The MOOC ran for the first time in May 2016 and, driven by consumer demand, it has run a further 10 times up until February 2020. It was ranked by learner ratings in the Top 100 MOOCs of All Time in 2019 [29]. The MOOC was purposefully designed to deliver a range of information about the relationship between nutrition, food and health and is delivered through the FutureLearn^TM^ platform. Engagement and interaction with learners is a key learning strategy in MOOC design so learner discussion is encouraged via forums throughout the six-week course [30]. There were 74 discussion forums within the MOOC inviting learners to discuss and comment on the topics presented in the course. These comments were either an original comment or responses to other participants’ comments. Providing comments was not mandatory for course participation. Data from the discussion forums of MOOC were collected and subjected to qualitative analysis and interpretation.

Data exploring the sources and vocational backgrounds learners shared were collected from the MOOC runs in May 2016, March 2017 and March 2018 and stored in Microsoft Excel (Microsoft Office Professional Plus 16) (Figure 1). The same two discussion forums from each of these runs (a total of six discussion forums) were selected for analysis. Two steps, “The Role Food Can Play in Prevention and Treatment” and “Foods and Prevention”, were chosen because they were the first two steps in the course where learners were least likely to be influenced by the remaining course content.

To explore social media use, data were collected from the discussions forums of the May 2016 run of the Food as Medicine MOOC (Figure 1). All comments containing terms relevant to social media were extracted for analysis. These data were analysed with the question ‘how are people using nutrition information found on social media’? Microsoft Access database was used for data management of the qualitative analysis.

### 2.2. Participants and Ethical Approval

The MOOC was specifically designed and targeted for the general public with no prior knowledge assumed or pre-requisite learning required. Provision of individual demographic data was not mandatory, with only country of origin collected for each individual but this information was not linked to learner discussion comments. All comments are in the public domain, with anyone able to join the MOOC and view the comments. Participants are notified by FutureLearn^TM^ when they sign up that comments made in courses may be used for research. Ethics approval by Monash University Human Research Ethics Committee (Ethics number: CF16/905).

### 2.3. Data Analysis

#### 2.3.1. Categorising Sources of Nutrition Information—Content Analysis

All comments from the six forums were examined and those identifying additional nutrition information sources were extracted for analysis. No identifying information of learners was collected.

Classification frameworks (Appendix B
Table A1 and Table A2) were developed by two researchers (M.A. and S.G.) by analysing the first 50 comments and identifying sources of information and authors’ vocational backgrounds. A further 100 comments were classified by M.A. and any new classifications were discussed with S.G. and the framework was adjusted accordingly. Classification saturation was reached at this stage. The remaining comments were then classified by M.A. according to the framework. Classification was conducted between September 2017 and January 2019. Any queries were discussed with S.G. and resolved. Provider types were classified according to their listed education background; a general classification was given (e.g., ‘tertiary educated professional: non-nutrition’), and then more detailed information on provider type was extracted (e.g., ‘Medical’). Each comment was analysed for the nutrition topic discussed, source of information (e.g., website, book), provider vocational background, and any additional notes that may be pertinent to the research (e.g., listing information that on a website about the provider such as ‘Physician and M.D.). Sources of information posted by learners were investigated by either clicking on the hyperlink used in the discussion forum or searching the information source on Google^TM^ where no link was provided. Details on provider backgrounds were determined by entering the provider name to Google and reviewing information on their internet profile. Analysis using descriptive statistics was conducted using Microsoft Excel.

#### 2.3.2. Learners Communicating Nutrition Information Found Online—Framework Analysis

To investigate how learners communicated nutrition information with each other, a framework analysis approach was adopted [31]. These data were analysed with the question ‘how are people using nutrition information found on social media’? A coding matrix was developed by two researchers, K.K. and M.A. after independent coding of 100 randomly selected posts which were discussed and compared by the 2 researchers. M.A. and K.K. each then coded all posts using the coding matrix. The researchers made note of any new codes identified in the analysis and were further discussed. K.K. and M.A. reduced codes to categories and cross checked the results analysed by the other researcher, discussing any discrepancies. A third researcher, S.G., then independently coded a random selection of posts, and compared these to results from K.K. and MA. Final themes were then decided by all three researchers (Appendix B
Table A3). Appendix A contains the COREQ (Consolidated criteria for Reporting Qualitative research) checklist for qualitative research.

### 2.4. Reflexivity

Before research began, the authors discussed any possible effects their background might have on the research; all researchers have training in nutrition science and dietetics, and work in a nutrition department at a university. M.A., S.G. and H.T. were involved in the development and delivery of the MOOC. The team discussed their assumptions and potential biases relating to their own educational and professional backgrounds prior to data analysis.

During the development of the classification framework, definitions for each were established to ensure a clear standard for classifications across multiple researchers and to reduce bias. There was high variability of qualifications and lack of standardised practice for people who classify themselves as ‘nutritionists’. M.A. and S.G. decided to use educational qualifications as the basis for classification of a provider’s vocational background to reduce any potential bias by the providers personal descriptions of themselves. Additionally, before research began, the researchers discussed experiences and observations regarding social media and dissemination of nutrition information, and how this may affect data analysis including potential feelings of frustration when presented with misinformation. An inductive approach to data analysis was adopted to minimise assumptions from the researchers influencing the outcomes of the research. Additionally, the quality and accuracy of information was not in the scope of this research to remove any potential prejudice against the providers and to more accurately classify their vocational backgrounds, and also to more accurately explore how learners are using information from social media.

## 3. Results

This section may be divided by subheadings. It should provide a concise and precise description of the experimental results, their interpretation as well as the experimental conclusions that can be drawn.

### 3.1. Learners Backgrounds

There were 62,144 (May 2016), 12,468 (March 2017), and 6738 (February 2018) people who registered for the three runs of the MOOC, from 197, 171 and 156 countries respectively. The top three countries’ learners that were from across all three MOOCs were Australia, United Kingdom and United States (Appendix B
Table A4). Countries’ data are based on the IP locations of joiners collected at the time of the learner enrolment in the course run as defined by Future Learn [23]. Learners ranged in age from 18 years and over across all three MOOCs (Appendix B
Table A4).

### 3.2. Where Did Learners Access Additional Nutrition Information?

From the six discussion forums analysed, 476 comments, made by 307 learners, contained links or other sources of nutrition information that learners were sharing. Figure 2 (and Appendix B
Table A5) summarises the sources of nutrition information learners accessed. A range of sources were referenced by learners, with the most common being online websites. Books, films and TV shows, social media and scientific journals were also sources of information referenced by learners. Interestingly, learners also referenced individual people as a source of information. Some nutrition information sources were identified as ‘non-specific reference to source’, meaning learners provided a reference to nutrition information without identifying the source type, e.g., a reference to the “Blood Type Diet” but did not reference whether this information was from a book, website, scientific paper, etc. Five sources were classified as ‘unidentifiable’ for reasons including not being in English or were unable to be located.

### 3.3. What were the Vocational Backgrounds of the Providers of the Nutrition Information Sources?

Figure 3 (and Appendix B
Table A6) summarises the range of vocational backgrounds of the providers. The most common providers of nutrition information were tertiary educated individuals without identifiable nutrition qualifications, including health care professionals, doctors, journalists and lawyers. The second most common group were information sources that had no identifiable provider information, such as no author listed. Nutrition information provided by nutrition professionals with recognised credentials such as Registered/Accredited Practising Dietitians or a university degree in nutrition science were not referenced often and were referenced a similar number of times to alternate practitioners such as naturopaths and kinesiologists. There were also providers who described themselves as being ‘self-taught’, with no identifiable education or training information. There were some sources that were classified as ‘unidentifiable’ due to factors including not being in English, inactive weblinks, etc.

The most common vocational background of providers, tertiary educated individuals without identifiable nutrition qualifications, was further categorised by their tertiary education and training. Thirty-seven percent of this group were medical doctors/had a medical degree, while three percent had a health care professional degree such as physiotherapy. Twenty-six percent of providers were categorised as ‘scientists’—individuals with a university tertiary degree in a science subject, but without health/medical training. Forty-three percent had professional training in areas unrelated to nutrition/medicine/science or health, such as journalism, economics, law and hospitality. Thirteen sources of nutrition information had multiple provider types from multiple vocational backgrounds, and, of these, nine included a nutrition expert.

### 3.4. How are People Using Nutrition Information?

Thematic analysis was undertaken to explore how learners communicated the nutrition information they found on social media.

Six hundred and thirty-nine posts, by 437 learners, contained references to social media and were included in the analysis. Four themes were identified: (i) teaching (advocating about what they have learned and/or reinforcing their beliefs), (ii) sharing information (with other learners and with their friends/followers on other platforms), (iii) learning for themselves (seeking/looking for information on social media), and (iv) providing their perception of social media.

#### 3.4.1. Learners as Teachers

Learners often assumed the role of the expert/teacher when sharing nutrition information from social media. There was a range of methods learners used to “teach” their fellow learners using imperative verbs. Some learners used strong, authoritative and directive language to teach information, instructing others to access social media links, groups or pages. In addition to providing information in a factual way, these learners also directed others to follow this information. For example, “Yes, lactofermented vegetable have been around forever. Have a look at youtube for one of the best at this called—the art of fermentation by XXXX—good luck” (Learner 3ae09304)

Others used social media references to present factual knowledge without any advice attached. These statements used a realistic tone and did not direct the learner to do anything, for example, “Research on Extra Virgin Olive Oil has shown that it contains properties such as polyphenols to help combat cardiovascular disease…. Hear more in this interview with Professor XXXXXXX from The XXXXXXXX University, USA—(YouTube link)” (Learner: bd2fe32a)

Learners also taught information by beginning posts with “I” statements, such as “I believe…” or “In my opinion…”, and then stating the piece of information or fact that they were attempting to teach, for example, “I disagree that large doses of vitamin C are not particularly useful, Ascorbic acid (vitamin C) can be lifesaving when given in large amounts and research has proven that fact (YouTube link)” (Learner: 053170fe)

Teaching also occurred through learners responding to individual learners’ posts with the purpose of providing targeted and individualised advice.

#### 3.4.2. Learners Sharing Information

Learners used links from social media to share information with others. Often, it was information the learner found interesting, relevant to the conversation or things that they thought other people might find interesting. These posts differed from teaching as the learners just shared information rather than assuming a teaching role and lacked imperative verbs. Sharing statements often included the words “this is interesting” or “this is useful”, and sometimes they also shared personal experiences or personal habits for example, “There are several websites and Facebook pages that deal with the subject of nutrition and health. XXXX Health on FB may interest you. Dr XXXX www.XXXXXX.com and XXXX XXXXXX www.XXXXXXXXX.com are online with many articles on nutrition etc. These are just a sample.” (Learner: 899228a0)

#### 3.4.3. Learners Using Social Media to Learn

Learners described how they use information from social media to learn information, get ideas or “inspiration” about health, nutrition or food. These statements lacked a ‘teaching’ component and often overlapped with sharing experiences of learning information from social media or sources of information on social media they regularly access for information that provide them with learning and motivation. Learners were inspired and learned from social media post, profiles or personalities, for example, “For a cough I recently made a mixture of pineapple, lemon, ginger, cayenne and manuka honey after coming across it on Pinterest.” (Learner: d854374a)

#### 3.4.4. Learners Perceptions of Social Media

Learners discussed their opinions and personal views of social media. Some learners resisted using social media as they found it difficult to follow or did not trust the information it provided. Some learners viewed the information available on social media as lacking evidence and a pervasive source of misinformation, for example, “With all the misinformation pedaled by unqualified celebrities and “insta-fit” social media types, it is more important than ever to lead a balanced and healthy lifestyle including a diet that is founded by evidenced based research.” (Learner: 7dcafc9c)

## 4. Discussion

This research explores for the first time the nutrition information-seeking behaviours of online learners in a nutrition-focused MOOC. Learners accessed nutrition information external to the MOOC from a range of sources, with the internet being the dominant resource. This finding is consistent with others which show the internet and social media as common sources of health information [9,11,32,33,34,35]. When analysing the provider backgrounds of the information learners sourced, nutrition expertise appeared to be underrepresented. The majority of provider backgrounds lacked documented nutrition-related expertise, or indeed any information on their vocational background.

This research found that only 5% of food and nutrition-related information shared by learners was written by nutrition professionals, such as Registered Dietitians and Registered Nutritionists. Dietitians are experts in nutrition and dietetics due to the extensive evidence-based nutrition training, assessment and accreditation they undergo, and, from this, expect to be perceived as experts by the public and their credentials valued accordingly [36,37]. Additionally, dietetic and nutrition professional associations educate the public on the expertise and role of nutrition professionals [38,39]. A 2018 Australian media release reported that less than 3/10 people trusted information from social media and health bloggers, with 9/10 Australians trusting a dietitian for nutrition advice. [40] This contrasts with our findings which suggest people are more likely to access information provided by non-credentialed people on the internet. While the public may trust dietitians for nutrition information, ease and immediacy of access could be a stronger influencer for those seeking nutrition information [34]. By believing the general public mostly turn to dietitians (or dietitian-authored information online) due to their expertise, rather than bloggers or other nutrition information purveyors, there is a risk that nutrition professionals will remain complacent in the online space and continue to lose influence. Interestingly, while the MOOC was led by dietitians and health professionals, this did not seem to influence the sources of information learners shared with each other—they still shared sources from providers without clear nutrition expertise.

Convenience and cost drivers influence people’s information-seeking behaviour, with the internet and social media allowing easier and cheaper access than visiting a health care professional in person [34]. A qualitative study reviewing the quality and accuracy of online nutrition and cancer information found that approximately half (48%) of the 100 online articles included for review lacked any author information [41]. Not only are people accessing nutrition information with little understanding of the expertise of its authors, but the way people use information they find by sharing and using it to teach others may encourage the spread of misinformation. Interestingly, our research found no-one quoted popular celebrities such as movie/sports stars. However, some providers of health information may be considered celebrities in their own right by the general public [2,17,42].

Our research demonstrates learners in a MOOC are using nutrition information they find on social media to not only learn information, but also to influence fellow learners through sharing of information with some learners adopting a teaching role. Consumption of information has changed, and people do not just consume information from experts passively. Connectivism is a knowledge learning theory relevant for e-learning, including MOOC-style education and the Web 2.0. [43]. It helps explain the complicated networks the learners engage with due to technology, and where they utilize knowledge acquired from being connected, including their social media networks, fellow MOOC learners, and the MOOC content itself. This continuous acquisition and contribution of knowledge by individuals challenges the traditional expert teacher–novice student dichotomy and was observed in this research. Connectivist learning theory was clearly demonstrated in the learners’ forum conversations, with learners acting as teachers as well as recipients of each others’ shared nutrition information.

With the increase in popularity of accessing nutrition information through online and social media, nutrition professionals such as dietitians should consider increasing their online engagement around meaningful discussions regarding nutrition, food and health information. The Academy of Nutrition and Dietetics (AND) suggests social media is a tool for dietitians to communicate with and educate the public [44,45]. Nutrition professionals need to recognise learning practices and theory, such as connectivism, are evolving with technology. With the general public now having autonomy over their learning and choosing the networks they participate in and who they learn information from, learners need access to evidenced-based nutrition information that is attractive and meets their needs [46]. Nutrition professionals need to regularly engage in a range of networks and communities and tailor messages accordingly.

While nutrition professionals are highly competent at communicating one-on-one with patients/clients, [38,39,47] they may lack skills in optimising the potential of technology to reach wide audiences, or do not see the relevance or importance. This may also be driven by the cost-model which does not reimburse time spent on social media which could be seen as ‘free advice’ [48,49,50]. To effectively compete, the nutrition profession needs to develop online communication skills to grow their online presence and become a prominent voice in the public nutrition information arena [2].

Effectively interacting in a range of networks can increase the frequency of evidenced-based information that people see. The Truth Effect (Illusory Effect) suggests that being repeatedly exposed to information increases an individual’s view that the information is true, regardless of the reliability of the source [51,52]. Regularly contributing sound information to online nutrition discussions, may help combat misinformation [53]. This is important considering our research highlighted not only do people access information from providers with a range of vocational backgrounds, but also learning and teaching happens between people online.

Collaborating with other health professionals is also required to ensure high quality nutrition information reaches online audiences. Medical doctors and health care practitioners were commonly referenced by the learners in this research, highlighting the importance of nutrition education is incorporated strategically into health care and medicine curricula [54]. Collaboration between the nutrition profession and medical and other health professions has the potential to reach larger audiences, considering doctors are considered as one of the most trusted sources of nutrition information [55,56].

This research included participants from around the globe and observed their online behaviour without prompts to indicate preferred source of nutrition information. This unique opportunity enabled the observation of community-led information learning and sharing, allowing participants to cite nutrition information spontaneously. Limitations included that learners already had some form of interest in nutrition by enrolling in a university-run MOOC, and thus may value evidence-based nutrition more than the general public, potentially overestimating the already low numbers of RD/RN references. The sample size was small considering the millions of people using the internet for sourcing nutrition information, and so translating these findings to other contexts must be done with caution. While sample sizes can be hard to define when observing people interacting online, RWD is growing in recognition as an important component of health research and important to hear participant voices regardless of the sample size. This research identified sources of nutrition information shared by learners, not whether they follow the recommendations shared. Although tertiary education of nutrition providers was used to classify vocational backgrounds of authors, it was beyond the scope of this research to evaluate whether this information was accurate, or whether providers had gained evidenced-based nutrition expertise outside of a formal degree. This analysis provides a contemporary snapshot of nutrition information-seeking behaviour. However, due to the rapidly evolving nature of this space, these practices need to be continuously monitored so health professionals can adapt and respond.

## 5. Conclusions

Learners not only sought nutrition information from an evidence-based nutrition MOOC, but further accessed nutrition information from providers with a range of vocational backgrounds. Nutrition professionals were under-represented sources for nutrition information and advice, with nutrition information providers having unclear nutrition expertise. Consistent with connectivist learning theory, learners contributed their own sources of nutrition information to discussions, using their own knowledge networks to teach and share information. Nutrition professionals need to appreciate the breadth of information sources and consider how to optimally create online presence so as to become a preferred source of nutrition information and advice by the public. Effectively communicating evidenced-based nutrition information that appeals to the public is essential to protect against misinformation and promote nutrition in the digital age.

## Figures and Tables

**Figure 1 nutrients-12-00750-f001:**
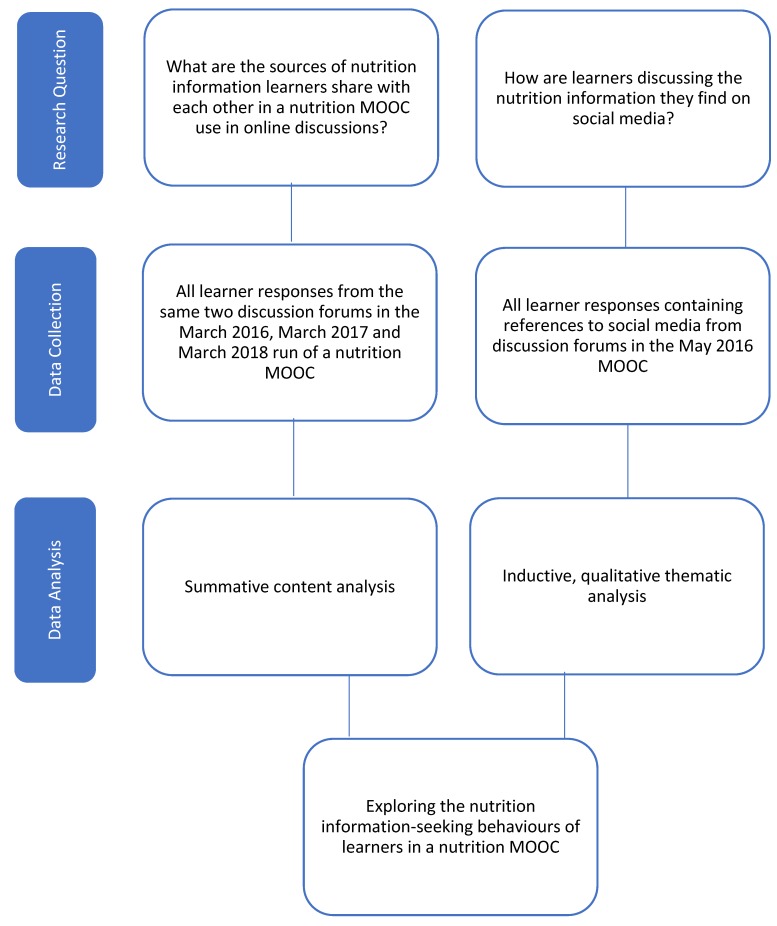
Overview of multi-methods approach and data collection of qualitative data.

**Figure 2 nutrients-12-00750-f002:**
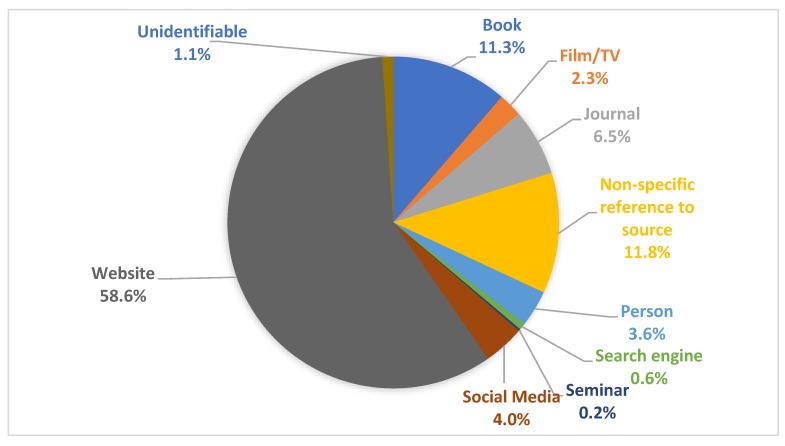
Results of sources of nutrition information.

**Figure 3 nutrients-12-00750-f003:**
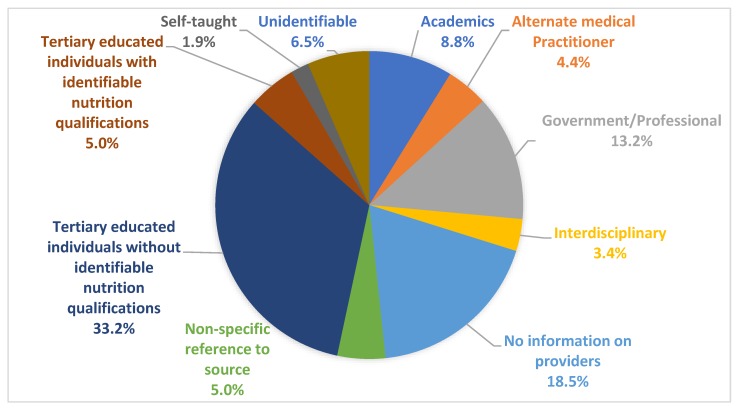
Results for provider vocational background.

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
