# Peer review of "Using the Internet: Nutrition Information-Seeking Behaviours of Lay People Enrolled in a Massive Online Nutrition Course"

_nutrients, 2020, doi:10.3390/nu12030750_

Round 1
Reviewer 1 Report
Dear Authors,
The topic discussed in the article is very important. The collected data required a lot of work, however, it seems that not all the goals that were established were achieved.
The three study questions were: 1. where people who are actively seeking nutrition information are accessing that information, 2. who they trust and 3. how they utilise the information they acquire and share. In my opinion, there is no answer to the second study question. You provide information only about “the vocational backgrounds of the providers of the nutrition information sources” – self reported or “if no information on author was found in comment itself, then a brief internet search was conducted”. This brings a high risk of bias and misclassification of educational and professional backgrounds.
Please reply to the following comments:
Introduction
- The introduction is too long. It tackles many topics, undoubtedly important, but seems unnecessary for a scientific article.
Material and Methods
- The figures A1, should be placed in the M&M section as a flow chart of the study. This figure needs refinement.
Results
- Number of countries (lines 208-209). In 2016 there were 62 144 people form 197 countries who registered for three runs of the MOCC. According to UN, there are 197 countries in the world, including Vatican, Palestine, Taiwan and Kosovo. Have all countries in the world joined the platform?
- A table, showing baseline demographic characteristics of the study group, which combine the data from supplementary tables 1 i 2, could be useful.
- Section 3.1. You have in total 476 comments containing links or other sources of nutrients information. How many per person?
- Section 3.2. The figure 1 needs refinement. I suggest adding percentages as in the figure 2.
- Section 3.2. lines 222-223: Four sources were removed from the final analysis for reasons including not being in English and were unable to be located. Please check if it is correct. Figure 1 does not provide percentages, but 4 answers seem to be a smaller percentage than is indicated in the chart.
- Section 3.3. Results are described twice (in the text and in the figure). Please include a synthetic description in the text. The figure 2 needs refinement.
- Section 3.4. The results are presented in a descriptive form. You divide all learners into 4 categories. Please present these data in a table.
Discussion
- Why do you state (lines 311-313) “Dietitians perceive themselves as experts in nutrition and dietetics due to…”? In my opinion they are experts in nutrition and dietetics.
- Please explain why (lines 322—323) “there is a risk nutrition professionals will remain complacent in the online space and continue to lose influence” when (lines 320-321) “the general public mostly turn to dietitians (or dietitian-authored information online) due to their expertise”?
- Please explain the term “a global sample of participant” (line 377).
- You claim that (lines 391-392): This analysis provides a contemporary snapshot of nutrition information-seeking behaviour,.. but the final result, that the majority of nutrition information sources learners reported were from websites… is nothing new.
Conclusions
- Conclusions are too general. The authors do not answer the study question.
In conclusion, the article requires major revisions before being published. Basic statistics would increase the value of the article.
Author Response
Response to Reviewer 1 Comments
Thank you for taking the time to review our manuscript – we appreciate your feedback and comments. Please see below for responses to your reviewer comments.
Point 1: The topic discussed in the article is very important. The collected data required a lot of work, however, it seems that not all the goals that were established were achieved.
The three study questions were: 1. where people who are actively seeking nutrition information are accessing that information, 2. who they trust and 3. how they utilise the information they acquire and share. In my opinion, there is no answer to the second study question. You provide information only about “the vocational backgrounds of the providers of the nutrition information sources” – self reported or “if no information on author was found in comment itself, then a brief internet search was conducted”. This brings a high risk of bias and misclassification of educational and professional backgrounds.
We agree 'who the public trust' was not adequately analysed. We have removed this from the manuscript – Line 113-114. Re vocational backgrounds question: We have changed the text to reflect the process we undertook – Lines 179-183. Change made “Sources of information posted by learners were investigated by either clicking on the hyperlink used in the discussion forum, or searching the information source on GoogleTM where no link was provided. Details on provider backgrounds were determined by entering the provider name to Google and reviewing information on their internet profile. Analysis using descriptive statistics was conducted using Microsoft Excel.” We agree there is subjectivity involved in classifying nutrition providers' backgrounds and have described the process used to minimise bias in the Reflexivity section. Line 200-218
Point 2: The introduction is too long. It tackles many topics, undoubtedly important, but seems unnecessary for a scientific article.
Agreed the introduction is lengthy. We covered multiple topics to help ensure we sufficiently introduced concepts that audiences with a nutrition background may not be familiar with, including MOOCs, and Real World Data. We have reduced the length of the introduction by removing paragraph on trust 77-85 and editing the overall introduction.
Point 3: The figures A1, should be placed in the M&M section as a flow chart of the study. This figure needs refinement.
Thank you for your feedback on figure A1. We have moved the figure to the M&M section (starting line 149) and refined it. It is now Figure 1 and lines 139 and 144 that refer to this figure have been amended.
Point 4: Number of countries (lines 208-209). In 2016 there were 62 144 people form 197 countries who registered for three runs of the MOCC. According to UN, there are 197 countries in the world, including Vatican, Palestine, Taiwan and Kosovo. Have all countries in the world joined the platform?
A table, showing baseline demographic characteristics of the study group, which combine the data from supplementary tables 1 i 2, could be useful.
Future Learn defines country data as " Country data is based on the IP locations of joiners collected at the time of their enrolment on this course run". This has been added to the manuscript. Supplementary tables 1 and 2 have been combined into one demographics table. This has been included in the Appendix as Table D. References to this table have been amended in the manuscript on lines 227-229.
Point 5: Section 3.1. You have in total 476 comments containing links or other sources of nutrients information. How many per person?
The 476 comments were by 307 learners – this is now added into the manuscript on line 231.
Point 6: Section 3.2. The figure 1 needs refinement. I suggest adding percentages as in the figure 2.
Thank you for highlighting this error. This has been amended with % now included – this figure is now Figure 2 and it has been refined.
Point 7: Section 3.2. lines 222-223: Four sources were removed from the final analysis for reasons including not being in English and were unable to be located. Please check if it is correct. Figure 1 does not provide percentages, but 4 answers seem to be a smaller percentage than is indicated in the chart.
Apologies again this was not clear due to the error in accidently omitting the % off Figure 1 due to choosing the wrong template in Excel. The numbers have been re-checked and an error has been amended that was in the text (there was a mistake as 4 resources were mentioned in original manuscript when it was actually 5). These numbers have been checked. The 5 sources that were removed equates to 1.1%, which is 5 out of 476 and the graph reflects this. We have now included tables with the results of Figure 2 and 3 in Appendix E.
Point 8: Section 3.3. Results are described twice (in the text and in the figure). Please include a synthetic description in the text. The figure 2 needs refinement.
Thank you for highlighting this - results have been re-written to include a synthetic description of the text – lines 232-245 and 249-262. Figure 2 has also been refined.
Point 9: Section 3.4. The results are presented in a descriptive form. You divide all learners into 4 categories. Please present these data in a table.
Thematic analysis results are most commonly presented as text in qualitative research which is why we have presented the data this way (Burnard P, Gill P, Stewart K, Treasure E, Chadwick B. Analysing and presenting qualitative data. British dental journal. 2008 Apr;204(8):429.). We have expanded the results table in appendix to include more detail to hopefully meet the needs of all types of readers. We have also added the following sentence to the manuscript "Thematic analysis was undertaken to explore how learners communicated the nutrition information they found on social media." Line 277-278
Point 10: Why do you state (lines 311-313) “Dietitians perceive themselves as experts in nutrition and dietetics due to…”? In my opinion they are experts in nutrition and dietetics.
We agree, in our opinion dietitians are the experts in nutrition and dietetics. However, not all professions (or the public) are aware of the training and scope of practice of dietitians, and do not necessarily perceive dietitians as the experts. We have changed the sentence to " Dietitians are experts in nutrition and dietetics due to the extensive evidence-based nutrition training, assessment and accreditation they undergo, and from this expect to be perceived as experts by the public and their credentials valued accordingly." Line 341-344
Point 11: Please explain why (lines 322—323) “there is a risk nutrition professionals will remain complacent in the online space and continue to lose influence” when (lines 320-321) “the general public mostly turn to dietitians (or dietitian-authored information online) due to their expertise”?
Thank you for highlighting this for clarification. 320-321- says 'by believing the public mostly turn to dietitians etc, nutrition professionals may remain complacent'. This sentence is after discussing a media release on the results of a survey that indicated that the public trust dietitians over bloggers. This part of the discussion explores while the media release reference suggests people trust a dietitian over bloggers, our research found people are turning to nutrition information from others providers, not necessarily dietitians. We wanted to highlight while people may trust dietitians, it doesn’t necessarily mean they seek information from them, and so nutrition professionals needs to be aware that just because they may be considered a trusted source of nutrition information, it doesn’t mean people will access information from them.
Point 12: Please explain the term “a global sample of participant” (line 377).
We have changed the sentence to "This research included a global sample of participants from around the globe and observed their online behaviour without prompts to indicate preferred source of nutrition information." Line 410
Point 13: You claim that (lines 391-392): This analysis provides a contemporary snapshot of nutrition information-seeking behaviour, but the final result, that the majority of nutrition information sources learners reported were from websites… is nothing new.
Thank you for your feedback on our analysis. We agree part of our analysis confirms previous results from research regarding people are increasingly using the internet for information. We believe though our final result goes beyond the fact the learners use the internet for seeking nutrition information, but describes the range of nutrition sources learner’s access, that credentials do not seem to be an important factor, and that learners use nutrition information to teach and share with others.
Point 14: Conclusions are too general. The authors do not answer the study question.
We have removed the reference to 'who people trust for nutrition information' from the introduction and so we have not addressed in the conclusion as it is not part of the research. We have also reviewed the conclusion and adjusted to more specifically answer the study question. Line 430-433.
Point 15: In conclusion, the article requires major revisions before being published. Basic statistics would increase the value of the article
Thank you again for reviewing our manuscript and providing suggestions for revisions. We appreciate these suggestions and have revised the manuscript to address these comments. While the content analysis uses basic statistics to present the data, a thematic analysis was chosen to explore how learners use nutrition information to explore the learner comments in greater depth and to develop themes and understand meaning. Using a qualitative approach to explore Real World Data allows for the learner 'voice' to be heard, explored with meaning e. We feel a quantitative approach wouldn't provide this level of depth for exploring the data. Morse, J. M. (2007). Qualitative Researchers Don’t Count. Qualitative Health Research, 17(3), 287–287.
Reviewer 2 Report
Interesting and fully current article.
The work is well presented and with an appropriate methodology.
The results are sufficient and well discussed. The limitations are well resolved and explained.
Reviewer 3 Report
This study is a very innovative one. The idea of analysing MOOC and RWD is very important, especially in this area of research. Therefore, I think the study is an important contribution to the existing literature. Nonetheless, some revisions are needed.
Abstract: not sure what is the methodology? In general, the abstract must be improved to grab the attention of the reader. This was not the case when I read it.
Although I am not a native speaker, the text reads difficult and non-scientifically in several occasions. Please improve, in particular the start of the introduction and the discussion.
Minor issues:
References 1-4 state what?
Reference 14 is incorrectly integrated in the text
Long sentence: 97-99. Try to split in 2 sentences.
Unclear what is meant in sentence 119-120. Please explain better.
There are no percentages in Figure 1, but there are in Figure 2. Please explain
Round 2
Reviewer 1 Report
I accept the authors' explanations. I have no more comments.